# Contrastive learning of cell state dynamics in response to perturbations

## Abstract

We introduce dynaCLR, a self-supervised framework for modeling cell and organelle dynamics via contrastive learning of representations of time-lapse datasets. Live cell imaging of cells and organelles is widely used to analyze cellular responses to perturbations. Supervised modeling of dynamic cell states encoded in 3D time-lapse data is laborious and prone to bias. dynaCLR leverages single-cell tracking and time-aware contrastive sampling to map images of cells at neighboring time points to neighboring embeddings. We illustrate the features and applications of dynaCLR with the following experiments: analyzing the kinetics of viral infection in human cells, detecting transient changes in cell morphology due to cell division, and mapping the dynamics of organelles due to viral infection. Temporally regularized embeddings computed with dynaCLR models enable efficient and quantitative annotation, classification, clustering, or interpretation of the cell states. The models reliably embed, i.e., generalize to, data from unseen experiments with different microscopes and imaging contrasts. Models trained with dynaCLR consistently achieve $> 95\%$ accuracy in mitosis and infection state classification, enable the detection of transient cell states and reliably embed unseen experiments. dynaCLR provides a flexible framework for comparative analysis of cell state dynamics due to perturbations, such as infection, gene knockouts, and drugs. We provide PyTorch-based implementations of the model training and inference pipeline and a napari plugin user interface for the visualization and annotation of trajectories of cells in the real space and the embedding space.

## 1 Introduction

Learning biologically interpretable representations of changes in the cell and organelle morphology captured by terabyte-scale time-lapse images is an open and essential problem. The changes in the functions of cells and organelles caused by perturbations, such as infection, modulation of gene expression, and drug treatments, alter the dynamics of cells and organelles. Detecting the morphological changes across perturbations with engineered features or human supervision is prone to bias and time-consuming. In contrast to supervised methods, self-supervised learning of visual representations of morphological dynamics of cells and organelles promises several advantages: it can enable statistically reliable measurements of morphological states, quantification of discovered cell states across many experiments, generalization across diverse datasets and conditions, and the discovery of causal relationships between the cellular responses and the perturbations. Self-supervised representation learning is a promising and scalable approach for analyzing cell state dynamics encountered in cell biology and drug discovery.

We report a self-supervised learning framework to analyze the dynamic cell states using multi-channel 3D time-lapse images. Unlike natural images, microscopy images have diverse channels, e.g., fluorescence channels that encode the distribution of specific biomolecules and phase channels that encode the material properties of cells and organelles. The distribution of biomolecules provides a rich yet complex encoding of the cell's functional states, such as cell division, replication of pathogens, immune response, and cell death. Cell states observed by single snapshots often appear highly variable due to the diverse responses of the cells to perturbations and the lack of temporal synchronization between cellular responses. The heterogeneity of cellular responses can be interpreted accurately by analyzing the dynamics of cell states.

We report the following methodological advances to enable quantitative analysis of cell and organelle dynamics in response to perturbations:

1. Self-supervised framework, dynaCLR, for mapping the 3D multi-channel images of single cells to a temporally regularized embedding space, where the distance between the embeddings reflects the temporal vicinity between the cell and organelle morphology. dynaCLR models generalize to data from imaging systems and cell types, making the learned embeddings useful for multiple downstream analyses.

2. Diverse downstream analyses of cells' morphological states from their dynaCLR embeddings: classification of the cell states in the embedding space with efficient annotations, measurement of the dynamics of the abundance of annotated cell states, and discovery of changes in cells and organelles due to perturbations.

3. A scalable PyTorch implementation for training models on GPU clusters and a napari plugin GUI for annotating cell states in real and embedding spaces.

4. 3D multi-channel time-lapse datasets of infected cells that include a ground truth reporter of infection at multiple spatial and temporal resolutions, suitable for assessing the generalization of models of cell dynamics.

The development of dynaCLR is driven by the problem of mapping the complex dynamics of cells and organelles in response to viral infection and cell cycle across multiple microscopes. We evaluate the accuracy of visual representation learned by our method using both computer vision and biologically relevant benchmarks, namely the distribution of distances in the embedding space across a dataset of tracked cells and the accuracy of the classification of the cell states using 3 hours of expert annotations.

## 2  BACKGROUND AND RELATED WORK

Self-supervised learning of visual representations of objects and scenes from videos (Wang and Gupta, 2015; Denton, 2017; Sermanet et al., 2018; Qian et al., 2021; Dave et al., 2021) has been an active area of computer vision. A recent comparison of generative and contrastive models for various prediction tasks by Liu et al. (2024a) suggests that both approaches can perform similarly for diverse computer vision tasks. An attractive feature of contrastive learning is that it can encode diverse prior knowledge about the relationships between the data points and the desired structure of the learned embeddings. The concept of contrastive learning was first introduced as dimensionality reduction via learning an invariant mapping (Hadsell et al., 2006). Since then, the idea of contrastive learning (Chen et al., 2020; He et al., 2020) has been applied for training foundational models of images and multimodal datasets (Radford et al., 2021). Contrastive optimization of the latent space of generative models has been reported to improve the expressivity of the model (Aneja et al., 2021).

In cell biology, self-supervised generative models that leverage time-lapse microscopy data have enabled analysis of immune response (Wu et al., 2022; Shannon et al., 2024), cell division (Soelistyo et al., 2022), segmentation (Gallusser et al., 2023), and plant phenotyping (Marin Zapata et al., 2021). Contrastive self-supervised models are also widely used in cell biology, for example, to learn diversity of mitochondrial shapes (Natekar et al., 2023) in response to perturbations, detect cell division (Zyss et al., 2024) and learn relationships between gene expression and images (Wang et al., 2024; Şenbabaoğlu et al., 2024).

Viruses exploit the host cell's machinery to produce new virions, reprogramming the structure and function of the organelles and the whole cell. For example, flaviviruses, such as Zika and Dengue, replicate on the Endoplasmic Reticulum (ER) derived membrane compartments(Verhaegen and Vermeire, 2024), leading to changes in its morphology, morphology of other organelles, and the morphology of the whole cell. The global impact of viral infection on cells and organelles has been studied using RNA-sequencing(Gutiérrez and Elena, 2022) and mass spectrometry(Bojkova et al., 2020; Hein et al., 2023). The -omics modalities have enabled the discovery of the changes in the molecular states of the cells due to perturbations. However, they do not directly report the dynamic remodeling of organelles and cells. Analyzing the cell and organelle dynamics in response to the perturbations requires 3D multi-channel (e.g., multiple fluorescent channels, phase only, or phase + fluorescence) imaging. Extracting biological insights from these datasets requires efficient models for

learning robust models that map 4D tensors in real space to embedding space that represents cell types and cell states of interest.

Earlier work on temporally regularized variational autoencoder (VAE) models (Wu et al., 2022) demonstrated that incorporating weak priors about the temporal smoothness of embeddings leads to models that generalize to unseen data. Contrastive learning is a flexible framework for encoding priors of similarity and dissimilarity between objects in a dataset across the dimensions of space, time, perturbations, and channels. Contrastive models also tend to be more parameter-efficient for discriminating cell phenotypes due to the absence of the decoder used in generative models, an essential feature for training models that embed 4D tensors. Considering the above trade-offs, we encode complex cell and organelle morphology with 3D multi-channel live cell imaging and decode the cell states using cell tracking and time-aware contrastive sampling.

Large-scale benchmark datasets of static images of perturbed cells (Chandrasekaran et al., 2023; Chen et al., 2024) are available. However, benchmark datasets of time-lapse images of perturbed cells are smaller in comparison (Edlund et al., 2021; Antonelli et al., 2023) due to the challenges of live imaging and annotations outlined earlier.

## 3 METHOD

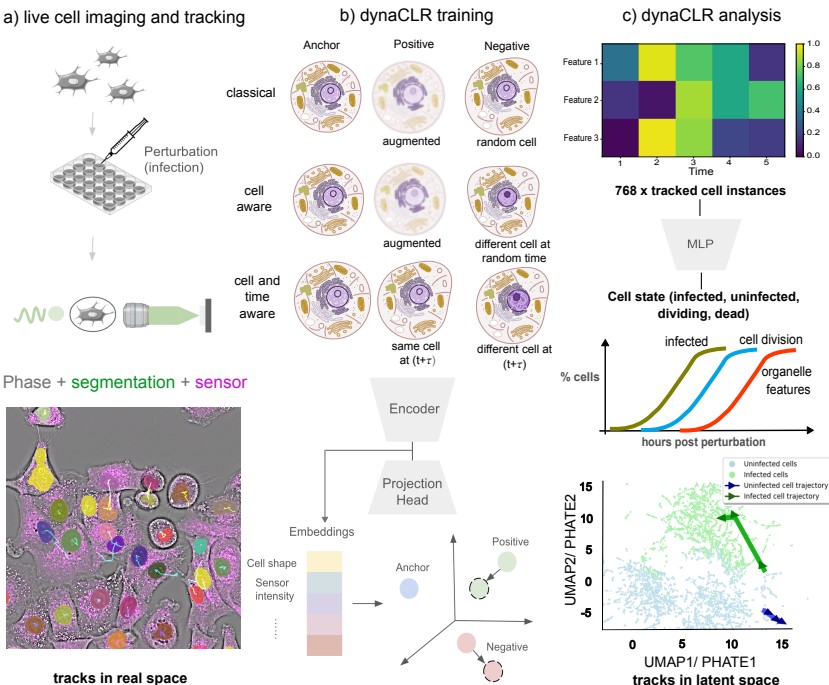

Figure 1: **Summary of dynaCLR:** (a) Live cells are perturbed, e.g., infected, and time-lapse images are acquired with correlative quantitative phase and fluorescence microscopy. Cell nuclei are virtually stained and tracked. (b) Contrastive loss with three different sampling strategies, classical (cell and time agnostic), cell-aware, and cell and time aware, is used to map multi-channel 3D volumes to embedding vectors. (c) Cell state dynamics are analyzed by classifying the cells from the embeddings, by measuring the abundance of cells in different states, and by joint interpretation of tracks in the latent and real spaces.

### 3.1 TIME-AWARE AND CELL-AWARE CONTRASTIVE SAMPLING

dynaCLR pipeline, illustrated in Figure 1, consists of two main tasks: a pretext task of learning temporally regularized embeddings and a task of identifying cell states from the embeddings of patches of single cells.

We embed 3D multi-channel patches of single cells $\mathbf{x} \in \mathbb{R}^{C \times Z \times Y \times X}$ subjected to different perturbations, including the intrinsic perturbation of time, where dimension $C$ represents channels. The cells $\mathbf{x}_i$ are tracked across time $t_1, t_2, \ldots, t_n$ as they transition through different states, e.g., division, infection, death, and innate immune response. dynaCLR models are trained with a set $\{\mathbf{x}_i(t)\}$ of tracks using different contrastive sampling strategies, where $i$ is track id, and not a batch index.

We evaluate three sampling strategies ( Figure 1b), summarized below, that can flexibly leverage morphological information encoded at diverse spatial and temporal resolutions in multiple channels.

We explore three sampling strategies:

- **time-agnostic and cell-agnostic sampling (classical):** This strategy is the same as classical contrastive sampling of natural images and does not use tracking. The pretext task embeds two augmented views of an anchor cell at a given time $\mathbf{x}_i(t)$ from all other images of the same or the other cells. The anchor $\mathbf{x}_a = \mathcal{A}_1[\mathbf{x}_i(t)]$ and positive $\mathbf{x}_p = \mathcal{A}_2[\mathbf{x}_i(t)]$ are created through augmentations $\mathcal{A}$, while negative examples are augmented views of random cells $\mathbf{x}_n = \mathcal{A}_3[\mathbf{x}_j]$ at random time points.

- **cell-aware sampling:** This strategy uses tracking to form the positive pairs from the images of the same cell and negative pairs from the images of distinct cells. Similar to the classical approach, the positive pairs $\{(\mathbf{x}_a, \mathbf{x}_p)\}$ are created from augmentations of the anchor image, but the negative pairs $\{(\mathbf{x}_a, \mathbf{x}_n), \mathbf{x}_a = \mathcal{A}_1[\mathbf{x}_i], \mathbf{x}_n = \mathcal{A}_3[\mathbf{x}_j], i \neq j\}$ are images of other cells at random times.

- **time-aware and cell-aware sampling:** Given an anchor image $\mathbf{x}_a = \mathcal{A}_1[\mathbf{x}_i(t)]$ at time $t$, this strategy uses tracking to sample an image of the same cell $\mathbf{x}_p = \mathcal{A}_2[\mathbf{x}_i(t + \tau)]$ as a positive example. An image of a different cell $x_n = \mathcal{A}_3[x_j(t + \tau)], i \neq j$ at time point $t + \tau$ is sampled as a negative example. The pretext task is to minimize the distance between embeddings of a given cell across the time interval $\tau$ and, simultaneously, maximize the distance between the embeddings of different cells over the same time interval. The time offset $\tau$ is a hyperparameter empirically chosen based on the temporal sampling rate and the time scale of the biological processes of interest. For the experiments in this paper, we report embeddings learned from datasets acquired with diverse sampling rates and multiple time intervals.

We optimize dynaCLR models with triplet loss (Weinberger et al., 2005) among the embeddings of the batches $\mathcal{B} = \{(\mathbf{x}_a, \mathbf{x}_p, \mathbf{x}_n)\}$ of anchor, positive, or negative triplets, as well as NT-Xent loss (Chen et al., 2020) among the embeddings of the batches $\mathcal{B} = \{(\mathbf{x}_a, \mathbf{x}_p)\}$ of anchor and positive pairs. For each positive pair in the batch, NT-Xent loss effectively treats all other samples as negative examples.

For downstream analysis( Figure 1c), we embed images using dynaCLR models and visualize them in low-dimensional space using PCA, UMAP, or PHATE (Moon et al., 2019), overlaying cell state annotations for infection and cell cycle. PHATE algorithm explicitly accounts for transitions in embedding space during dimensionality reduction and preserves continual progressions and branches better than UMAP. We use PHATE and UMAP transforms only for visualization and not for measuring distances or classification.

## 3.2 Data, annotations, and metrics

We explore the features and evaluate the generalization of dynaCLR models with two datasets: an already published 2D time-lapse dataset that encodes cell cycle dynamics and a new 3D time-lapse dataset that encodes infection and cell cycle.

### 3.2.1 Annotated label-free images (ALFI) of HeLa, RPE1, and U2OS cells

We use DIC movies of three cell types from this dataset (Antonelli et al., 2023) in which bounding boxes of a subset of cells were tracked with human annotation. The time points of the tracks are also annotated with cell cycle state (mitosis vs interphase). DIC is a widely used label-free microscopy method. Note that *label-free* in the context of microscopy implies the absence of fluorescent labeling of cells and not the absence of human annotations of cell states. We evaluate the ability of dynaCLR models to discriminate mitosis and interphase stages of the cell cycle that generalize to unseen cell types. The training set consists of

unperturbed HeLa and RPE1 cells, and an independent test set consists of perturbed and unperturbed U2OS cells, all acquired in 2D with a time resolution of 7 min. We further assess the generalization of the cell cycle embedding model to quantitative phase images of infected A549 cells described next.

### 3.2.2 LABEL-FREE AND FLUORESCENCE IMAGES OF DENGUE VIRUS INFECTED A549 CELLS

A549 cells infected with live Dengue virus were used as a model system for self-supervised discovery of cell states(Figure 1a). We acquired 3D time-lapse datasets with two fluorescent channels and quantitative phase channel as described in Section A.2 on two distinct microscopes: spinning disk confocal with phase imaging channel (Guo et al., 2020) and a light-sheet microscope with phase imaging channel (Ivanov et al., 2024). We use virtual staining of nuclei (Liu et al., 2024b) and multi-hypothesis tracking with Ultrack (Bragantini et al., 2024).

The dataset used for model training was acquired with a 30 minute temporal resolution on the light-sheet microscope. We used a subset of fields of view (FOVs) from the experiment, including cells infected with a multiplicity of infection (MOI) of 5 viruses/cell and mock-infected wells, for training. We used two independent test experiments to evaluate the generalization of models: a) a dataset acquired with a 30-minute time resolution on the spinning disk confocal and b) a dataset acquired with a 10-minute time resolution on the light-sheet microscope. Both independent test datasets contained mock and MOI 5 conditions.

We trained models with two channels: the phase channel that encodes the global responses of the cell and a fluorescent channel that either encodes the state of infection or the response of an organelle. An infection reporter was imaged in a fluorescence channel. The infection reporter construct used in this study consists of a fluorescent protein (mCherry) with a nuclear localization signal (NLS) and ER anchor peptide separated by a cleavage site recognized by a viral protein(Pahmeier et al., 2021). This expressed protein is localized to the outer membrane of the ER under normal conditions. Upon infection with the Dengue virus, which expresses the viral protease, mCherry-NLS is freed from the ER anchor and translocates to the nucleus. Thus, nuclear localization of viral sensor provides *experimental annotation* of viral infection.

Cell division or mitosis is a significant event in the cell cycle that causes significant changes in cell morphology. Mitosis is marked by the condensation of chromosomes and the rounding of cells as genetic material separates, visible in the phase images (Guo et al., 2020). During mitosis, the sensor is localized in the nucleus whether or not the cell is infected, which confounds the detection of the infection state from the snapshot. In contrast to the transient changes in morphology seen during mitosis, cells that become infected remain infected over time, as captured in both label-free and fluorescence channels. Learning the image embedding from neighboring time points enables the disambiguation of cell states in such cases.

### 3.2.3 ANNOTATIONS OF INFECTION AND CELL DIVISION

We evaluated the trained model with a manually curated test set with reliable annotations for cell division and infection states. We validated the annotations and predictions by overlaying them on the projected embeddings. We also tested the model on independent test data to assess its generalization to new data.

Infection state annotation was based on manually revised annotations from a 2D-Unet model (Liu et al., 2023), adapted for semantic segmentation and three-class classification using weighted cross-entropy loss. The model classified patches of pixels into three categories: background (0), uninfected nuclei (1), and infected nuclei (2). The annotations were proofread and edited using a custom napari (Chiu et al., 2022) plugin. The proofreading of the semantic segmentation model's predictions was necessary due to the inability to accurately capture late infection stages and cell death, as these states often resulted in a loss of fluorescence signal and altered cell morphology.

Cell division is captured from cell tracking by Ultrack (Bragantini et al., 2024) and revised manually. The cell division is indicated by a parent track splitting into two daughter tracks with the same parent track IDs. The last time-point of the parent track is considered the

Table 1: Linear classification accuracy and macro-averaged F1 score for interphase vs. mitosis classification. Results also include dynamic range and smoothness metrics as defined in Appendix A.1

| Experiments | Accuracy (%) | F1 (%) | Dynamic Range | Smoothness |
|---|---|---|---|---|
| Cell & Time Aware ($\tau = 0$) | 97.0 | 96.6 | 0.31 | 0.13 |
| Cell & Time Aware ($\tau = 7$) | 97.7 | 97.4 | 0.39 | 0.12 |
| Cell & Time Aware ($\tau = 21$) | 97.7 | 97.4 | 0.55 | 0.16 |
| Cell & Time Aware ($\tau = 28$) | 97.8 | 97.5 | 0.55 | 0.17 |
| **Cell & Time Aware ($\tau = 56$)** | **97.8** | **97.6** | **0.56** | **0.17** |
| Cell & Time Aware ($\tau = 70$) | 97.1 | 96.8 | 0.51 | 0.16 |
| Cell & Time Aware ($\tau = 91$) | 97.4 | 97.0 | 0.50 | 0.17 |
| Classical (no tracking) | 96.4 | 95.9 | 0.27 | 0.14 |
| Cell Aware | 98.0 | 97.7 | 0.46 | 0.14 |

division event. The human annotator proofread and corrected the cell division events through visual inspection of the tracks in Ultrack GUI.

We developed the napari plugin to link the dynamics of cells in the embedding space (visualized via UMAP, PHATE, or PCA projections) with dynamics in the real space in multiple channels(Appendix Figure 1). This plugin enables inspection of the evolution of the cell and organelle phenotypes and interactive annotations of clusters of phenotypes.

### 3.3 Metrics

Similar to other contrastive learning approaches, we evaluate the dynaCLR models with linear classification accuracy. We use two binary classification tasks (mitotic vs interphase cells and infected vs uninfected cells) to assess the discrimination of the cell states in the embedding space. Half of the annotated test data was used to train a logistic regression classifier from the embeddings, and the other half was used to evaluate the classification accuracy using % accuracy and F1-scores. At the time of inference and linear classification, it is not necessary to use tracks of cells.

We evaluate the dynamic range and smoothness of the embedding space by analyzing temporal changes in the distance (either Euclidean or cosine similarity) of tracks of embeddings $\mathbf{z}_i(t) \in \mathbb{R}^E$ as described in Section A.1.

## 4 Experiments

### 4.1 Temporal regularization via time-aware contrastive sampling

Using the ALFI dataset imaged with high time resolution (7 min), we analyzed the temporal regularization of embeddings and classification of cell division state in the embedding space. Models with classical, cell-aware, and cell & time-aware contrastive sampling strategies were trained using HeLa and RPE1 cell types that were not perturbed by drugs using triplet loss. The model architecture, training, and data augmentations are described in the Section A.3, Section A.4, and Table 2.

When unseen cells (U2OS cells with and without drug treatment) were embedded with trained models, the displacement of embeddings of a specific cell (Figure 2a) and mean displacement of the embeddings over the test dataset (Appendix Figure 2) showed that cell and time-aware sampling increased the dynamic range of the embeddings relative to the classical and cell-aware contrastive sampling.

Notably, the dynamic range and smoothness are maximized for the time intervals ($\tau$) of 28 to 56 minutes(Table 1). The displacement of embeddings reduce for longer time intervals of 70 and 91 min (Appendix Figure 2). PHATE visualization of the embeddings of U2OS cells (Figure 2b and Appendix Figure 3) with classical and cell & time-aware contrastive sampling show that time-aware sampling results in an embedding space that reflects the transitions and branches in the cell shape due to cell division events. The time-aware sampling (56-min interval) results in smoother embedding space (smaller local distances) with a larger dynamic range (larger global distances) compared to the classical or cell-aware sampling for this dataset and hyperparameters of the loss function.

Further, the linear classification accuracy Table 1 is maximized by the model trained with a time interval of 56 minutes, which is close to the typical duration of 1 hour of mitosis. These experiments show that time-aware contrastive sampling improves the smoothness and dynamic range of the embeddings relative to classical and cell-aware sampling, and the time-interval hyperparameter enables tuning of the dynamic range per dataset.

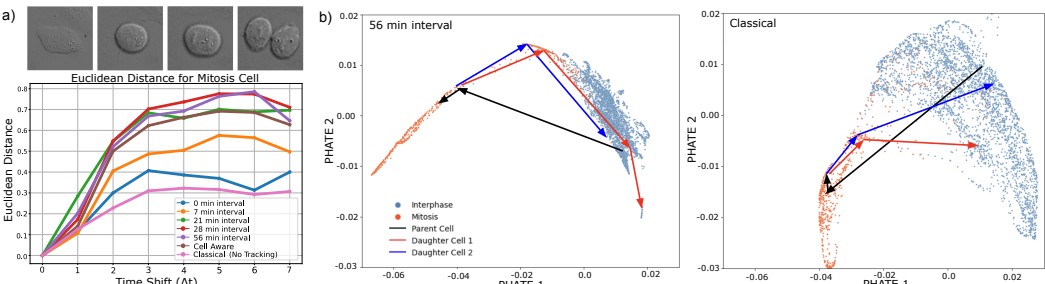

Figure 2: **Time-regularized embedding:** a) An example track of a mitotic U2OS cell. The displacement of the embedding of the cell at different time intervals (Equation 1) is shown for multiple contrastive sampling strategies (see legend). The dynamic range, i.e., the peak displacement across all $\Delta t$ is the highest for cell & time aware sampling with time interval $\tau = 56$min. b) Trajectories of a dividing cell are visualized with PHATE as the cell transitions from the interphase cluster (parent, black track) to the mitosis cluster, followed by the daughter cells (red and blue tracks) re-entering interphase clusters.

This property is helpful for the downstream analysis of cell dynamics. It is also a more challenging pretext task that helps the model learn a richer representation of the cell state due to the increased biological variation between positive pairs. This approach improves embedding tolerance to batch effects like photobleaching, as demonstrated in (Figure 3d) and f), where our model trained on non-photobleached data generalizes to photobleached test datasets.

Next, we evaluate the effect of the choice of the contrastive loss function (triplet vs. NT-Xent) on the structure of the embedding space using the data from infected A549 cells. The models were trained using data acquired with a time resolution of 30 minutes on the light-sheet microscope and evaluated with an annotated independent test set (Section 3.2.3) acquired on the confocal microscope under similar experimental conditions. The PHATE visualization of the embeddings of the annotated test set (Appendix Figure 4b) and the distribution of distances (Appendix Figure 4c) show that NT-Xent loss leads to slightly better dynamic range and smoothness in embeddings.

Further computational experiments with infected A549 cells and triplet loss were performed to benchmark the accuracy of infection classification with cell- and time-aware contrastive sampling. The metrics are shown in Table 3. The above data show that the classification of the infection state of cells from dynaCLR embeddings consistently achieves ≈ 95% accuracy compared to the semantic segmentation baseline that achieves ≈ 80% accuracy, given the same amount of annotations. Further, time-aware contrastive sampling improves learned embeddings' rank (possible shape modes) as seen from Appendix Figure 5.

Taken together, the above results establish that the cell-aware and time-aware sampling strategy improves the continuity and dynamic range of the embedding space independent of the choice of contrastive loss.

## 4.2 DYNACLR MODELS GENERALIZE ACROSS EXPERIMENTS AND MICROSCOPES

In addition to the efficiency of annotation, an important advantage of self-supervised learning with biologically informed pretext tasks is that such models generalize to unseen data. In other words, they are robust to confounding factors in data collection. We conducted three sets of computational experiments to assess the generalization of dynaCLR models for the downstream analysis tasks of cell state classification, infection classification, and discovering organelle remodeling due to infection. Cell division or mitosis is a rare event characterized by large changes in the cell morphology, resulting in two daughter cells. The cell divisions

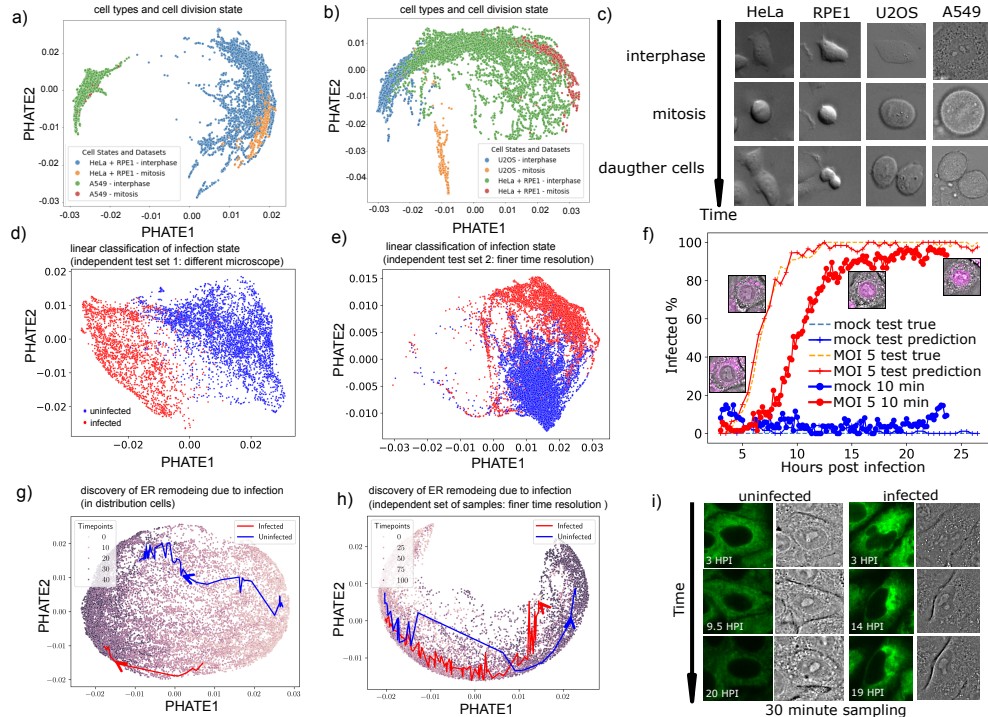

Figure 3: **dynaCLR models generalize across cell types and microscopes**: (a and b) PHATE maps from model trained with DIC timelapse images of HeLa and RPE1 cells clusters mitotic and interphase in other cell types: A549 imaged with quantitative phase and U2OS imaged with DIC. (c) Example images of HeLa, RPE1, U2OS, and A549 cells progressing from interphase to mitosis, dividing into daughter cells. (d and e) PHATE maps showing infection states clustering on predictions on images from a different microscope and different time sampling. (f) The plot of the percentage of infected cells over hours post-infection shows a similar trend between ground truth and prediction on data from a different microscope, as well as with data with different temporal sampling. (g and h) PHATE maps show the dynamic state of cells with the SEC61 organelle marker evolving over time. (i) Phase and fluorescence images of cells from mock and infected conditions, organelle remodeling evolving over time in the infected cell.

are distinctly visible in the label-free imaging channel and are independently detected by the tracking algorithm. Many perturbations, including infection, modulate the rate of cell division.

### 4.2.1 DYNAMICS OF DIVIDING CELLS IN EMBEDDING SPACE

We evaluated the joint embedding spaces of the cell types used for training the cell division model (HeLa and RPE1 from ALFI datasets) with two independently imaged cell types: U2OS cells from ALFI imaged with DIC(Figure 3a) and A549 cells from our dataset imaged with quantitative phase (Figure 3b).

Notably, the embeddings separated cell types as well as cell states according to the similarity of their shapes (Figure 3c). The embeddings of similar cell types and division states are contiguous in the embedding space as visualized by PHATE. Further, the trajectory of a cell in the embedding space is robust to occasional errors in tracking that may arise in dense cell cultures (Appendix Figure 6). We think the dynaCLR cell division model generalizes to unseen cell types because of the more challenging pretext task of discriminating the cell shape in the future.

### 4.2.2 Dynamics of infected and dividing cells in embedding space

We used phase and sensor channels of light-sheet movies acquired with 30-minute time resolution to train a time-aware dynaCLR model and evaluated with two independent test sets acquired with 30-minute time resolution on a confocal microscope and 10-minute time resolution on the light-sheet microscope. The embeddings were classified with a linear classifier trained only on the small number of annotations from the confocal test dataset. Figure 3d-e shows the PHATE visualization of the embeddings of the test datasets with an overlay of the predicted class. The computed percentage of infected cells from half of the test data closely matched the infection percentages derived from human-revised infection dynamics in both mock and MOI 5 conditions, with the number of infected cells rising exponentially and plateauing at 12 HPI. A similar trend was observed in the independent test data, where infections plateaued at 15 HPI (Figure 3f). Thus, the infection classification model trained with dynaCLR framework demonstrated robust generalization across microscopes and multiple experiments.

We evaluated the possibility of detecting cell division using just the phase channel. dynaCLR embeddings change measurably as cells transition from interphase to mitosis, as seen from the tracks in the UMAP space (Appendix Figure 7), particularly in models trained solely with the phase channel and incorporating temporal regularization (Appendix Figure 7c). Smooth transitions and tight clustering of division events are also evident in models trained with both channels (Appendix Figure 7e). In contrast, the cell trajectories exhibit random walks in models trained without temporal regularization (Appendix Figure 7d), and clustering is less distinct when using both channels (Appendix Figure 7f).

### 4.2.3 Organelle remodeling during infection

Viral infection causes restructuring of organelles, such as the condensation of the endoplasmic reticulum (ER) where replication sites are established(Cortese et al., 2020). The range of organelle responses to specific perturbations can be challenging to define a priori. By tracking cells in the learned representation space, we can correlate the observed organelle remodeling with the other cell states, such as infection and cell cycle.

We trained a time-aware dynaCLR model to explore these relationships using a 30-minute temporal resolution dataset using the organelle fluorescence channel and phase. We evaluated its ability to highlight the ER remodeling due to infection within and out of distribution using the 30-minute and 10-minute time-resolution datasets, respectively (Figure 3g-h). Structural changes of the ER are shown in (Figure 3i and Appendix Figure 8), showcasing progressive ER condensation through infection.

### 4.3 Explanations of the cell state classification

We now explore explanations of the phenotypes learned by dynaCLR model trained with infected cells using two approaches: a) rank correlation between principal components of learned embeddings and engineered features, and b) feature attribution methods.

We observed robust clustering of infection states through principal component (PC) analysis (Figure 4a). The cell patches along the PC axes were examined to interpret the principal components (Figure 4b-c). The PCs were correlated with the image features identified from human inspection. The first few PCs were correlated with features such as the radial intensity profile, area of the fluorescence of the infection sensor, interquartile range (IQR), and standard deviation of the values in the phase channel, likely due to the change in density distribution in cells during the progression of infection.

To explain which patterns in the input images influence the classification of cell states, we use Captum's implementation (Kokhlikyan et al.) of occlusion perturbation (Zeiler and Fergus, 2014). These attribution methods identify pixels in the input space that are most important for classifying the cell state. Classification heads for infection and division states are attached to the same encoder trained with phase and sensor channels and time-and-cell-aware sampling. Class attribution is then computed with occlusion perturbation ( Figure 4) for binary classification tasks of classifying the infection and cell division. Through self-supervised training, the encoder learns meaningful features that describe cell state dynamics, such as viral sensor translocation for infection and chromosome condensation for division.

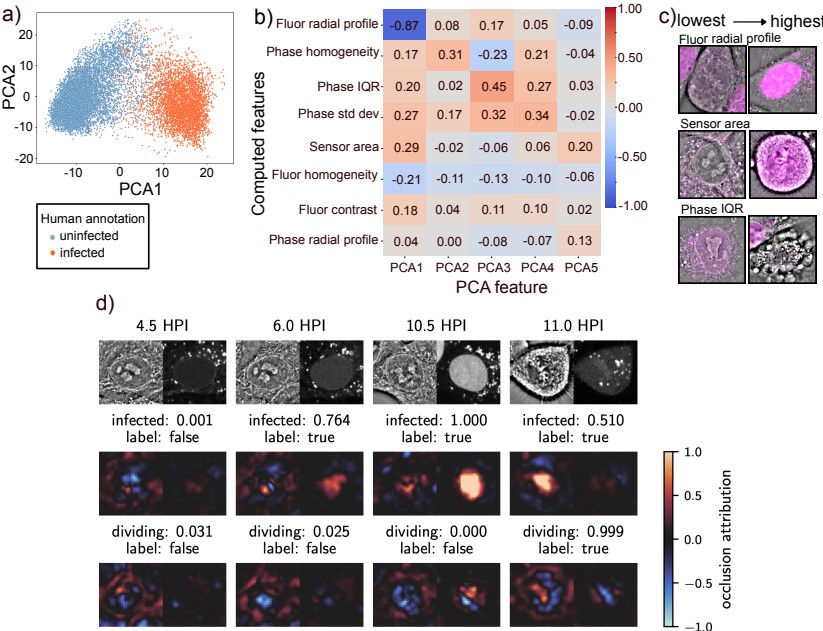

Figure 4: **Explanation of embeddings learned by dynaCLR:** (a) The first two principal components (PCs) of the test data, colored by human annotation of the state of infection, demonstrate that the largest variation in the embeddings is due to infection. (b) The rank correlation between computed features (Y-axis) and the first five PCs (X-axis) assigns meaning to the learned features. (c) The visual inspection of cells with the lowest and highest values along the principal component axes confirms the interpretation of PCs. (d) Occlusion attribution of a cell undergoing infection and division: The first row shows a center slice of the input images at different time points, the second row shows attribution with an infection classification head, and the third row shows attribution with a division classification head. The titles show the predicted probability and true class.

## 5 CONCLUSION AND FUTURE WORK

The above results show that tracking cell dynamics and time-aware contrastive learning (dynaCLR) leads to representations of cell and organelle morphology that encode smoothness of changes in morphology that enables multiple downstream analyses: discovery of abundant and rare cell and organelle states, classification of cell states with efficient annotations, and quantification of cell state dynamics.

The above datasets, computational experiments, and analyses open the following avenues for learning cell state dynamics:

- Learning a foundational model representing diverse cell states in many cell types is an exciting area of research. Leveraging the time and perturbation-aware contrastive sampling of time-lapse imaging datasets is a potential strategy for training such a model.

- Our current models pair label-free and fluorescence channels to encode cell and organelle states. Training channel-adaptive models that provide biologically interpretable embeddings of datasets with heterogeneous channels is an exciting future direction.

## 6 DATA AND CODE AVAILABILITY

We have attached an anonymized code to this submission. After the double-blind review, the code, models, weights, and datasets will be available via public repositories on GitHub, Huggingface, and Bioimage Archive.

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

# A  APPENDIX

## A.1  METRICS: DISPLACEMENT, DYNAMIC RANGE, SMOOTHNESS

To characterize the random walk of the cell in the embedding space, we compute the squared Euclidean displacement ($D$) between time points $t$ and $t + \Delta t$ after normalizing the embeddings at each time point, as follows:

$$D_i(t, \Delta t) = \left\| \frac{\mathbf{z}_i(t)}{\|\mathbf{z}_i(t)\|_2} - \frac{\mathbf{z}(t + \Delta t)}{\mathbf{z}(t + \Delta t)\|_2} \right\|_2 \tag{1}$$

The displacements relative to the start of individual tracks are used to compute the mean displacement and the standard deviation over a population of tracks.

We assess the effect of the contrastive sampling method and the loss functions on the temporal regularization of the embedding space using dynamic range and smoothness metrics: The smoothness is defined as the opposite of the average distance between adjacent frames in the dataset, i.e., $1 - \mu[\{D_i(t, t + 1) : \forall i, \forall t\}]$. Smoothness represents the temporal continuity of embeddings. The dynamic range represents the maximum range occupied by the test dataset in the embedding space. It is defined as the difference between the maximum and minimum displacements (Equation 1) across all time shifts ($\Delta t$).

## A.2  IMAGE ACQUISITION AND PREPROCESSING

We acquired 5D images (time series of 3D volumes of phase and fluorescence images) of A549 cells infected with live Dengue viruses at a multiplicity of infection (MOI) of 5. The infected and uninfected cells were imaged for over 24 hours in multi-well plates - the wells without the virus are called mock-infected wells. We acquired the data as OME-TIFF stacks using MicroMananger(Edelstein et al., 2010) and converted it to OME-Zarr format using iohub for high-performance handling of large image data.

The phase images are obtained from deconvolution of the brightfield images captured with Köhler illumination(Guo et al., 2020). The phase images represent the density variation in cells and inform the model on the overall changes in the morphology of cell(Guo et al., 2020; Wu et al., 2022; Ivanov et al., 2024) during events like infection and cell division, as well as the location of organelles like cell nucleus(Liu et al., 2024b) and ER relative to the whole cell.

## A.3  MODEL ARCHITECTURE AND TRAINING

The model architecture has three main components: a spatial projection stem, an encoder backbone, and a multi-layer perceptron (MLP) head. The stem begins with a convolution layer with a kernel size of $(5, 4, 4)$ and a stride of $(5, 4, 4)$, followed by a reshaping operation. This reshaping maps the down-sampled axial dimension to channels, efficiently projecting the anisotropic 3D input into a 2D feature map for encoding. The backbone is adapted from the ConvNeXt Tiny architecture (Liu et al., 2022), using ImageNet pre-trained weights (noa, 2024). The stem and head modules from ConvNeXt are removed, and the backbone outputs a 768-dimensional embedding vector $\mathbf{h}$. The 768-dimensional vector $\mathbf{h}$ is mapped into a lower 32-dimensional space through a 2-layer MLP head, which helps speed up training (Chen et al., 2020).

The models are trained with a mini-batch size of 256, using the AdamW optimizer (Loshchilov and Hutter, 2019), and a learning rate of $2 \times 10^{-5}$. The triplet margin objective is used with a margin of 0.5.

## A.4  SAMPLING AND AUGMENTATION OF PATCHES OF SINGLE CELLS

3D imaging volumes are cropped around the centroids of the tracking nodes to form single-cell patches. We normalize the input image to reduce variability from experimental conditions. For the sensor fluorescence channel, we rescale the image so that the median intensity is 0, and the 99th percentile intensity is 1. This normalization is more robust to extreme highlights in the fluorescence image, as well as variation in background fluorescence levels. The quantitative phase channel is normalized so that each field-of-view (FOV) has zero mean and unit standard deviation. The phase image is already normalized during reconstruction (Guo et al., 2020), and this extra standardization step ensures proper input numerical range for the model. We use a larger initial crop to ensure no padding is included in the final

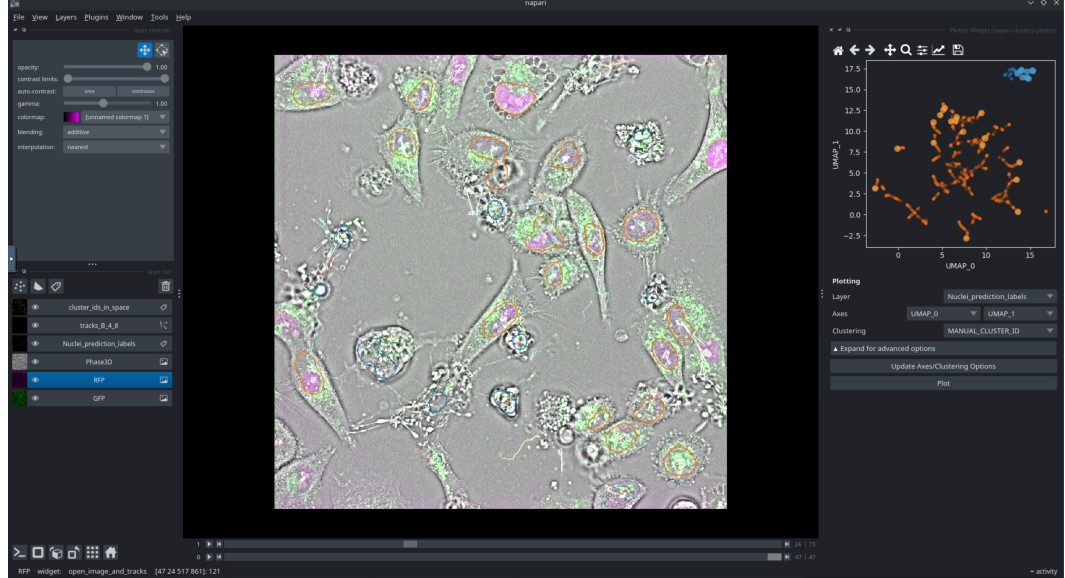

Appendix Figure 1: A napari workflow for interactive exploration and annotations. We developed a napari plugin to load images, tracks, and learned features. Then napari-clusters-plotter is used for the interactive annotation of the cell dynamics in latent space with reference to the morphological changes in real space.

input patch after spatial augmentations. We apply extensive augmentations (Table 2) at training time to simulate variations induced by the imaging system and other non-biological conditions. The input patch size after augmentations is $[15 \times 128 \times 128]$, which is optimal for reducing the influence from background and neighboring cells while focusing on the peri-nuclear region of the cell, where the majority of infection-related changes such as sensor relocalization and ER remodeling are captured.

Table 2: Augmentations applied to image patches. Parameters are supplied to respective MONAI (Cardoso et al., 2022) transforms, where $\alpha$ denotes scaling factor, $\theta$ denotes rotation (radians), $s$ denotes shearing, $\gamma$ denotes gamma value, $\sigma$ denotes standard deviation of the Gaussian distribution, and $p$ denotes the probability of applying the random transform.

| TYPE | PARAMETERS |
|---|---|
| Random Spatial Scaling | $\alpha_x, \alpha_y \in [-0.3, 0.3], p = 0.8$ |
| Random Rotation | $\theta_z \in [0, \pi], p = 0.8$ |
| Random Shearing | $s_x, s_y \in [0, 0.01]$ |
| Random Adjust Contrast | $\gamma \in [0.8, 1.2], p = 0.5$ |
| Random Intensity Scaling | $\alpha \in [-0.5, 0.5], p_{\text{Phase}} = 0.5, p_{\text{RFP}} = 0.7$ |
| Random Gaussian Smoothing | $\sigma_x, \sigma_y \in [0.25, 0.75]$ |
| Random Gaussian Noise | $\sigma_{\text{Phase}} \in [0, 0.2], \sigma_{\text{RFP}} \in [0, 0.5], p = 0.5$ |

Table 3: Accuracy of classification of infection state with supervised semantic segmentation model and self-supervised contrastive models trained with phase and sensor channels

| Model | Accuracy (%) |
|---|---|
| Supervised semantic segmentation model | 83 |
| Contrastive w/ Linear classifier (Cell & Time Aware sampling) | 97.5 |
| Contrastive w/ Linear classifier (Cell Aware sampling) | 97.9 |
| Contrastive w/ Linear classifier (No tracking) | 98.8 |

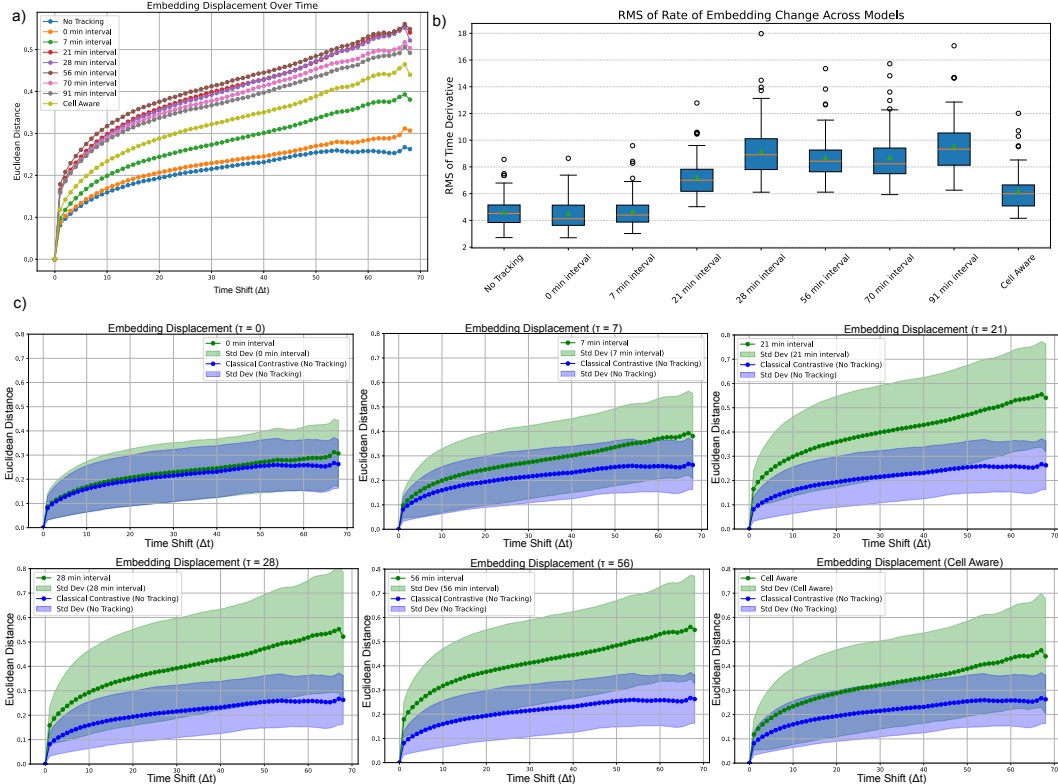

Appendix Figure 2: **Evaluation of embedding displacement and temporal smoothness across sampling strategies.** (a) Mean embedding displacement over time ($\Delta t$) for various sampling strategies, showing the highest dynamic range at $\tau = 56$ and a decrease for higher values. (b) Root Mean Square (RMS) of the rate of embedding change (time derivatives) across different temporal sampling strategies, highlighting the temporal smoothness achieved by each model. (c) Individual mean embedding displacement plots over time for each sampling strategy compared to the classical (no tracking) approach. Cell and time-aware strategies consistently achieve a higher dynamic range than classical models showing their effectiveness in capturing temporal relationships in embedding space.

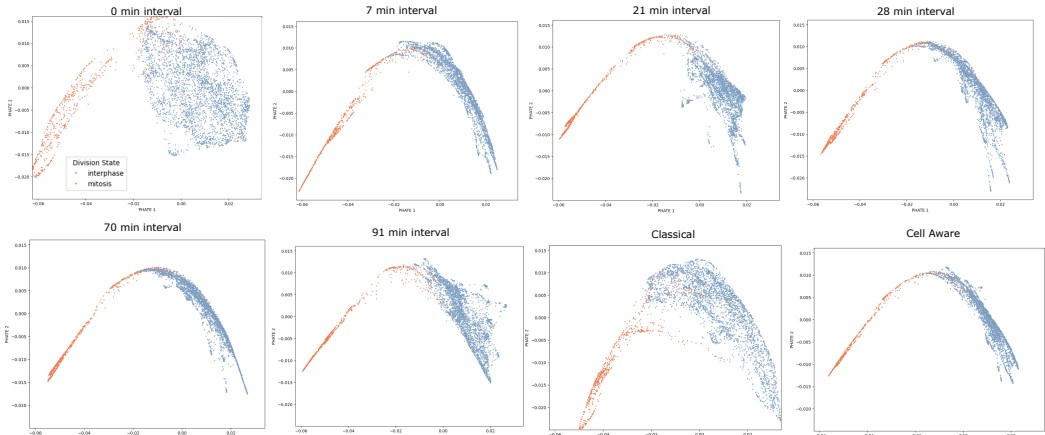

Appendix Figure 3: **PHATE embeddings of cells under different sampling strategies.** These panels show the PHATE embeddings for cells across various time shifts ($\tau$) and sampling strategies. Time-aware sampling (e.g., 7 min, 21 min, 28 min intervals) results in embeddings where similar cell states are closer and more continuous in embedding space, potentially reflecting improved temporal alignment. In contrast, classical sampling exhibits greater scattering and discontinuity in cell state trajectories. The embedding continuity highlights the ability of time-aware sampling to better preserve temporal relationships between cell states compared to classical methods.

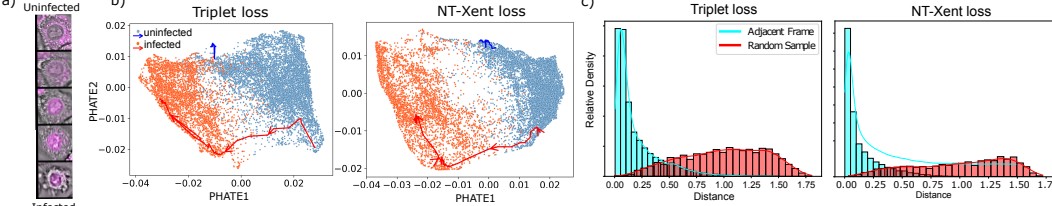

Appendix Figure 4: **Effect of loss function on learned embeddings:** (a) Progression of infection in a cell from the top to the bottom image shown by translocation of infection sensor. (b) PHATE maps of Dengue-infected cells colored by ground truth infection state from models with triplet loss (left) and NT-Xent loss (right) show similar clustering and cell trajectories over time. (c) The plot shows the measurement of smoothness. The cyan histogram measures the dissimilarity of one track with respect to the subsequent timepoint for all tracks, and the red distribution is for random tracks.

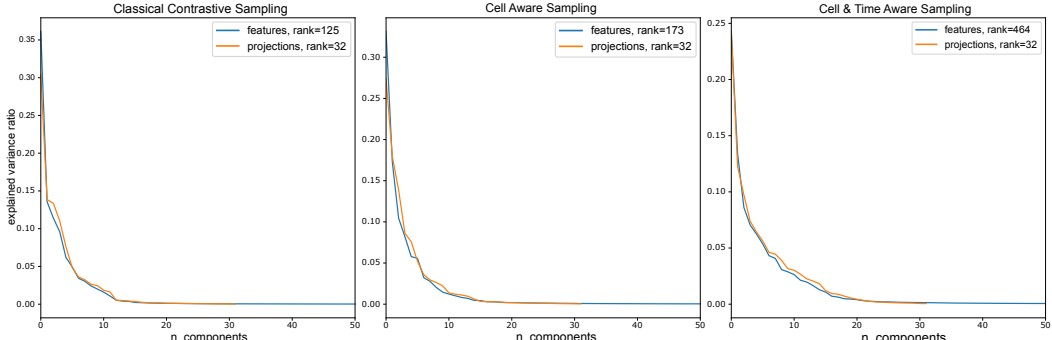

Appendix Figure 5: Comparison of explained variance ratios across three different sampling strategies: 1) **Classical Contrastive Sampling (No Tracking)**: Rank of features is 125, and projections is 32, showing that the classical approach without tracking captures less variance, with the explained variance ratio dropping steeply within the first 10 components. 2) **Cell Aware Sampling**: Rank of features is 173 and projections is 32, showing a slightly broader variance explained by initial components, indicating improved variance capture when cells are tracked. 3) **Cell and Time Aware Sampling**: Rank of features is 464 and projections is 32, indicating the highest rank and broader variance explained across components, which suggests that incorporating both cell and time information improves the embedding space's representational richness.

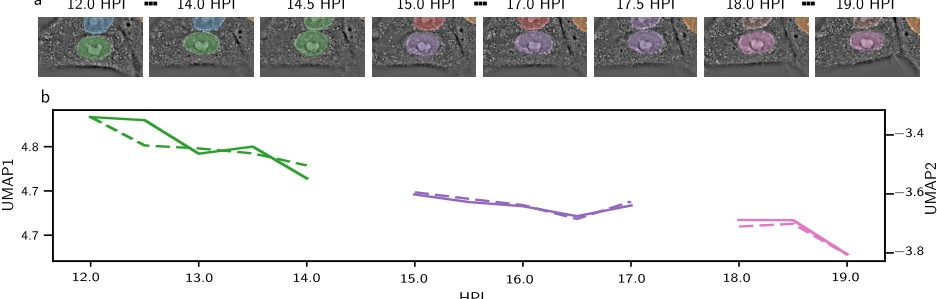

Appendix Figure 6: **dynaCLR embeddings are smooth even when tracking is erroneous:** (a) snapshots of a cell and its tracking labels over time. Note that the false fusion in 14.5 and 17.5 HPI frames caused subsequent false division and identity jump of the cell. (b) UMAP components 1 (solid line) and 2 (dashed line) over time for the falsely assigned tracks. The gaps correspond to false fusion events which shifts the centroid of the track towards the edge of the FOV, resulting in invalid patches. The UMAP components are smoothly transitioning over time, even though they are assigned to different tracks.

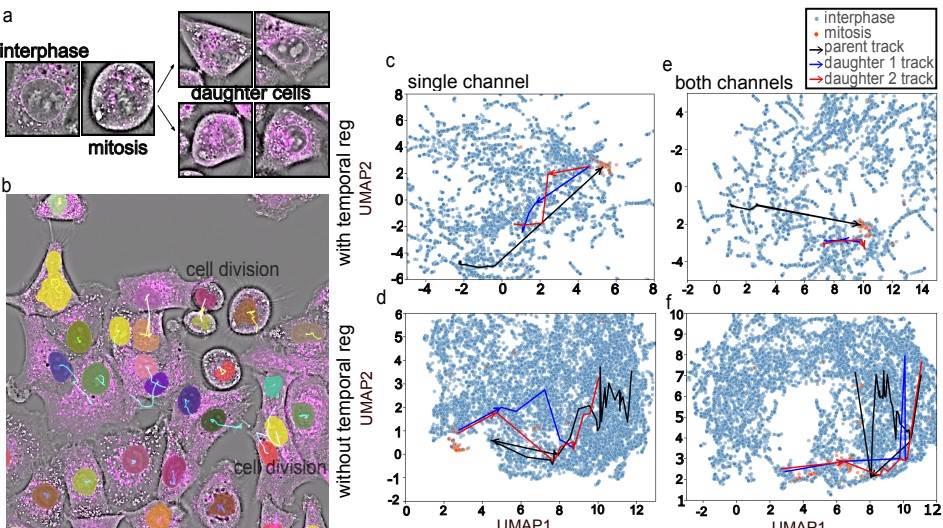

Appendix Figure 7: **Detection of rare events, e.g., cell division**:(a) The morphology of the cell changes over time during the transition between interphase and mitosis. (b) Ultrack tracks the cell over time and captures mitosis. White tracks indicate cell divisions. (c–f) The trajectory of one parent cell (black track) dividing into two daughter cells (blue and red tracks) overlaid on the UMAP from models using phase channel and a combination of phase and sensor fluorescence channels, and with and without temporal regularization, illustrates that temporal regularization leads to smooth trajectories and better clustering with just the phase channel.

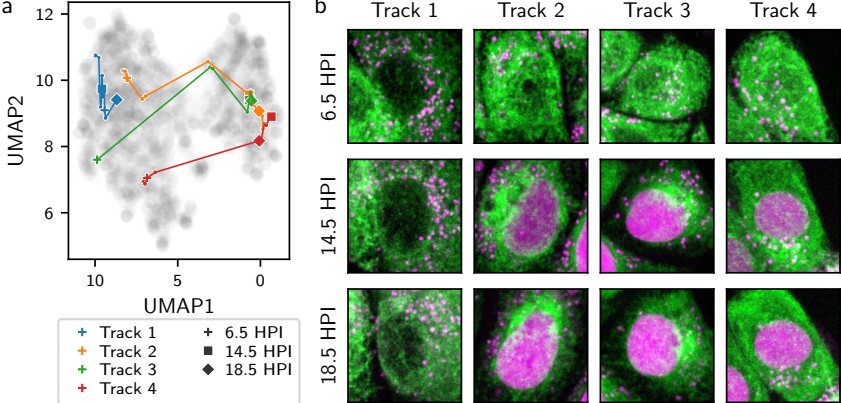

Appendix Figure 8: **Learned representation of the phase and sensor channels help exploration of organelle remodeling during infection.** (a) UMAP of learned features computed for mock and Dengue infected cells in the independent test dataset where the ER of cells is labeled with a fluorescent protein (SEC61-GFP). 1 track from the mock well and tracks 2-4 from the Dengue infected well are highlighted. Cells other than the example tracks are marked in gray. (b) Snapshots from example tracks in (a), showing max-intensity projection of ER (green) and the viral sensor (magenta). In some of the infected cells (tracks 2 and 3), ER forms transient condensation.

