# OpenReview forum: "Contrastive learning of cell state dynamics in response to perturbations"
_ICLR.cc/2025/Conference — Submitted to ICLR 2025_

### Official Review · Reviewer_UA5v · 2024-11-02

**Soundness:** 2
**Presentation:** 3
**Contribution:** 2
**Rating:** 3
**Confidence:** 4

**Summary:**

The paper describes an application of contrastive learning on a microscopy dataset to model single cell dynamics. The method introduces time-aware sampling to the traditional contrastive learning methodology by making use of cell-tracking. Three downstream applications of the learned embeddings are shown which demonstrate how the method can be used to analyze cell-state dynamics in response to perturbations. In addition, experiments showing generalization and interpretability of embeddings are performed.

**Strengths:**

1.	The paper addresses an interesting topic where self-supervised learning has the potential to be extremely beneficial. Getting annotations for large amounts of microscopy data on a single-cell level is difficult and there is a broad push in the field towards the kind of self-supervised approach described here, so this work is well placed.
2.	The utilization of cell tracking to improve self-supervised learning is an interesting idea; cell-identity detection is potentially a useful proxy task that could aid in learning good representations.
3.	The biological setup and experiments are well thought out and the dataset would be beneficial to researchers as a benchmark, given simple readouts like infection rate.
4.	The addition of generalization experiments (though only on one additional set of data) provides some grounding to the results – without this section I don’t think the paper has much weight. I would encourage the authors to extend this to more datasets if possible.

**Weaknesses:**

1.	In general, this is written like a paper in a scientific journal rather than a machine learning conference. While it is important to provide the necessary biological context, the paper lacks the mathematical discourse required to be suitable in a machine-learning setting. For example, could you add a theoretical or mathematical description of ideas such as ‘smoothness’ or ‘richness’ of the latent space that you have mentioned?
2.	I’m not quite sure that the quantitative results match up with the claims made. For example, performance scores in Table 1 are better for the ‘classical’ contrastive learning approach as opposed to the ‘cell-aware’ and ‘time-aware’ approaches. The arguments made based on the Euclidean distance plots in Figure 2 are strenuous, and I don’t agree with the qualitative conclusions made about time-regularized contrastive learning being better than the classical approach from these results (I will add more on this in the next points). Overall, I don’t think the contributions here are significant enough and the arguments don’t hold enough weight for me to accept this paper.
3.	The whole ‘temporal continuity’ argument seems a little counterintuitive to me. The ‘time-aware’ loss function is designed to encourage the model to ignore differences at the time-scale of the time hyperparameter Tau, but you expect the model to maintain temporal differences at time-scales t > Tau. I think Tau needs to be much smaller than the time-scale of your expected changes for this to work. If you expect changes in cells to happen over a few hours, the hyperparameter Tau being 30 minutes doesn’t make sense – your model is just going to smooth out all temporal information. I know you are limited by the fact that you can’t take images, say, every 5 minutes, but the current setup doesn’t make sense to me.
4.	Just to add to the previous argument – in your current setup, it matters how long you train your model for, since overtraining could completely smooth out all temporal information. I don’t even know how you would design experiments to determine what a good number of epochs to train is without having an idea of how much temporal smoothness is ‘correct’. For example, in the limit of an infinite number of epochs under this loss function, all temporal information would be removed from your model. I just don’t see how you could both encourage temporal smoothness while maintaining a temporally faithful embedding when your data has a temporal resolution of 30 minutes over 24 hours.
5.	In the case of ‘time and cell-aware sampling’, how do you ensure that the model is robust to imaging factors like brightness or sensor noise? The point of such augmentations in the classical contrastive learning case is to teach the model to ignore spurious confounders which may actually show up in your real data – but the ‘time and cell-aware’ model never learns to do this. Is this model actually good at generalizing? What happens of your imaging conditions change a little? Would this model still work? I think many questions need to be answered here.
6.	I think the novelty proposed here lacks conceptual correctness in my view. With traditional ways of using temporal information in self-supervised learning, like predicting the temporal order of randomly flipped images, it makes sense how these would lead to a temporally unbiased and meaningful embedding space. My arguments in the points above highlight my concerns on why the proposed method may not be doing the same.
7.	Without actual ground truth annotations of fine-grained temporality, the only way to assess the quality of the embeddings is through the results on downstream temporal tasks. However, the classical contrastive sampling variant seems to perform better on the downstream infection classification task, which tells me that the classical method is leading to better quality embeddings than the proposed method.

**Questions:**

1.	I’m a little bit confused about the quantitative results. Specifically, you write “Therefore, we evaluate the accuracy of visual representation learned by our method using a biologically relevant benchmark: accuracy of the classification of the cell states with 3 hours of expert annotations. We compare our method with two baseline methods: fully supervised time-agnostic semantic segmentation of the infection state and time-agnostic contrastive learning. Compared to the ≈ 80% accuracy achieved by the supervised model and 60 − 65% accuracy achieved by the time-agnostic contrastive learning, DynaCLR models consistently achieve ≈ 95% accuracy.” I only see results in Table 1 that show classical contrastive sampling having an accuracy of 98.8%. Can you clarify where the results of the supervised model are and why there is a mismatch between the claimed values of 60-65% and the values shown in the table?
2.	When you say, “For the experiments in this paper, we set τ = 30 min, as we found this captures significant cell changes while maintaining temporal continuity without over-sampling.” Does this mean you tested with different τ values? If you did any experiments on this it would be good to include those, and would contribute to my concerns in the weaknesses section on temporal continuity and temporal faithfulness.
3.	In Section 3.3 you claim, “We compute (2) and (3) to evaluate the temporal evolution of the embeddings as a gradual, steady increase signifies strong temporal smoothness. In both figure 2 a) and 2 b), cell and time-aware sampling shows a smooth rising curve with minimal fluctuations. The classical contrastive method exhibits a rapid initial increase followed by plateaus, while the cell-aware approach shows intermittent fluctuations.” I’m not sure I see in Figure 2 what is described here. It seems like the curves for Euclidean distance are pretty similar between the different strategies. In fact, in the UMAP version it seems like the cell and time-aware version is most fluctuating (whether UMAP is an appropriate space in which to perform this is another question, since it is known to not maintain global structure). Can you please elaborate on this? I’m also unsure what you mean by “richness” of representation in Figure 2C and how you determined this.
4.	Can you elaborate the results of the integrated gradients based interpretability in Figure A1? What exactly is the model focusing on in the two channels? Can you provide evidence that this is relevant to the output task?
5.	You say “We observe a clear transition of cell states from interphase to mitosis as we follow the cells in UMAP space, particularly in models trained solely with the phase channel and incorporating temporal regularization” I kind of see what you’re trying to say but it’s still very unclear. Can you provide a mathematical comparison of the two models and show that one model is ‘smoother’ than the other over all cell division events?

---

> ### Author Response · Authors · 2024-11-26
>
> Thank you for the thorough review. We address your feedback below:
>
> **Weaknesses:**
>
> Weakness #1:
>
> Thank you for this input. In this revision, we have introduced mathematical notation that describes the contrastive sampling strategy. We also use the precise term (dynamic range) instead of richness. We have developed metrics (see Appendix A.1) of dynamic range and smoothness based on intra-track and inter-track distances in the embedding space to rank the models.
>
> Weakness #2, #5, #6, #7:
>
> In this work, we prioritize the models that achieve higher dynamic range, smoothness, and classification accuracy with efficient use of human annotations. The goal is to visualize and identify the evolution of multiple biological states in the embedding space in response to the perturbations. The new results (Fig. 2, Appendix Figures 2 & 3, and Table 1) convincingly show that increasing the dynamic range of embeddings with cell- and time-aware models leads to consistently better classification accuracy than the classical contrastive models.  We found that the previous embeddings of infected cells were affected by the photobleaching of the infection sensor on the confocal microscope, which might have compromised the accuracy of cell and time-aware models. So we use our light-sheet imaging system to collect training and test data at two different time resolutions (see section. 3.2.2).  We soften the claim for infection classification that self-supervised models are more annotation efficient than dense semantic segmentation models.
>
> Weakness #3, #4, #6, #7:
>
> Thank you for the question. An intuitive argument for the time-aware positive and negative sampling is that it promotes intra-track smoothness and inter-track discrimination.  The large shape space of randomly chosen negative samples at t+Tau in a given batch should prevent the model from ignoring the differences in shapes. To evaluate this, we have trained several cell cycle models using public (ALFI) data acquired with 7-minute time resolution and multiple time intervals (Fig. 2, Appendix Figures 2 & 3). We also visualize the structure of embeddings with PHATE (https://github.com/KrishnaswamyLab/PHATE) rather than UMAP, as PHATE better preserves the continuity of embedding space in low-dimensional projections. We have also applied our infection model to a 10-minute temporally sampled dataset (figure 3, panel e and f) that shows that the model trained with the 30-minute dataset enables the analysis of the infection dynamics in data acquired with 10-minute time resolution.
>
> **Questions:**
>
> Question #1:
>
> The supervised method was used as a step to bootstrap infection annotation, which a human annotator further corrected to serve as the ground truth of infection for testing the DynaCLR infection state model. The results from the supervised model are added to Table 3 in the current iteration. 60-65% accuracy was achieved for a self-supervised model trained to predict infection state from just phase channel, which we have dropped from the table to avoid confusion. We have edited the table and text to clarify the data.
>
> Question #2:
>
> In this iteration, we have included evaluating the optimum temporal sampling for smooth predictions using multiple models trained with a publically available DIC dataset with various temporal sampling. You can find the new results in Figure 2 and Appendix Figure 2 and 3. We have also demonstrated the application of the model trained with 30-minute sampled data on a 10-minute sampled dataset in Figure 3, panel e.
>
> Question #3:
>
> We’ve dropped the results where we computed this metric in UMAP space since, as you’ve pointed out, UMAP does not maintain a global structure. Instead, we now report metrics in the full embedding space (see Figure 2 and Appendix Figures 2 and 3). We have also started using PHATE to better display the global structure of embeddings in low-dimensional projections.  We think that the definition of dynamic range and smoothness in (Appendix A.1), the data in Table 1, and the discussion of the experiment in Section 4.1 will clarify the concepts of dynamic range and smoothness.   Please ask us a follow-up question if they do not.
>
> Question #4:
>
> Thanks for pointing out the lack of “interpretability” of this specific figure. We found that occlusion analysis provides smoother feature attribution than the integrated gradients. We now report feature attribution with occlusion analysis in Figure 4d.
>
> Question #5:
>
> We've mathematically defined smoothness in appendix  A.1. We benchmark smoothness on the ALFI division dataset which was trained on the DIC channel. DIC is similar to phase as both are label-free imaging systems that shows density-based morphological features in cells. In table 1, you can see that time & cell aware sampling performs better on this smoothness metric with tau’s 21 and greater compared to the classical and cell aware sampling that do not leverage temporal information.

---

> > ### Author Response · Authors · 2024-12-03
> >
> > Dear reviewer UA5v,
> > We'd appreciate any follow up questions and revised review soon.

---

### Official Review · Reviewer_aWZ5 · 2024-11-03

**Soundness:** 3
**Presentation:** 3
**Contribution:** 2
**Rating:** 5
**Confidence:** 3

**Summary:**

In this paper, the authors propose a framework for modeling cell dynamics in response to perturbations. They suggested several downstream tasks for the learned representation, such as the analysis of viral infection kinetics in human cells, detecting transient changes in cell morphology, and mapping organelle dynamics due to viral infection. Furthermore, they reported that the proposed framework achieves an accuracy of over 95% for infection state classification, outperforming the supervised setting.

**Strengths:**

- The paper is well-written, following a clear structure that enhances readability and comprehension.
- The learned representation was used in several dowstream tasks.

**Weaknesses:**

- The technical novelty of the paper is limited, as DynaCLR may be considered a straightforward application of the triplet loss with varied sampling strategies.
-  Some figures (Fig. 2, Fig. 3, and Fig. 6) are not thoroughly explained; a more detailed description would enhance clarity.

**Questions:**

- The choice of the triplet loss is not justified. Why is this loss more suitable for this task than alternatives like NT-Xent or InfoNCE?
- In Section 4.1.1, could you clarify ithe origin of the independent test data?
- In Section 4.2, the smooth transitions and tight clustering of division events are not immediately evident in Figure 4; additional support for these claims would be beneficial.
- In Section 4.3, it remains unclear how the referenced figures demonstrate that "the encoder learns meaningful features that describe cell dynamics."
- Figure 6 would benefit from a more detailed explanation.

---

> ### Author Response · Authors · 2024-11-26
>
> Thank you for the review and for acknowledging the strength of our work. We address your feedback below:
>
> **Weaknesses:**
> ```
> The technical novelty of the paper is limited, as DynaCLR may be considered a straightforward application of the triplet loss with varied sampling strategies.
> ```
> We want to state that the novelty of the work is the following:
> * The first use of time-aware contrastive sampling to learn embeddings from 4D tensors.
> * Application to diverse biological datasets in which cell states are labeled by a human or via an experimental fluorescent marker.
> * We now use precise mathematical notation to clarify the contrastive sampling strategy, the metric of dynamic range in embedding space, and the metric of smoothness in embedding space.
> ```
> Some figures (Fig. 2, Fig. 3, and Fig. 6) are not thoroughly explained; a more detailed description would enhance clarity.
> ```
> Thank you for the comment. We have reorganized the paper to enhance clarity, recreated the figures with more datasets and experiments, and added clear descriptions in the text.
>
> **Questions:**
> ```
> The choice of the triplet loss is not justified. Why is this loss more suitable for this task than alternatives like NT-Xent or InfoNCE?
> ```
> In this iteration of the paper, we have experimented with both triplet loss and NT-Xent loss (Appendix Figure 4) and show that we obtained similar results using either loss function. We thank the reviewer for the question, which improved the quality of the work.
> ```
> In Section 4.1.1, could you clarify the origin of the independent test data?
> ```
> We have changed the independent datasets used and used two new independent datasets instead for infection state classification experiments. Please refer to section 3.2.2 for details on the two independent datasets for infection. We have also included results from the publicly available cell cycle DIC data, ALFI. The independent dataset used is a different cell type, U2OS, whereas the model was trained on images of HeLa and RPE1 cell lines.
> ```
> In Section 4.2, the smooth transitions and tight clustering of division events are not immediately evident in Figure 4; additional support for these claims would be beneficial.
> ```
> The data used in earlier iteration did not have enough division events, thus being a rare event and class imbalance issue. So, in this iteration, we have used ALFI, a publicly available dataset with human annotation of stages of the cell cycle for model training. Please refer to Figures 3, panels a and b, to see the results on clustering with cell division state and the model's generalizability to other cell types.
> ```
> In Section 4.3, it remains unclear how the referenced figures demonstrate that "the encoder learns meaningful features that describe cell dynamics."
> ```
> The paper has been reorganized, and these results have been revised to provide more insightful results.
> ```
> Figure 6 would benefit from a more detailed explanation.
> ```
> We have generated results from a new, improved model with a new dataset and added an explanation in the text of this figure. Please see Figure 4, panels g, h, and I.

---

> > ### Author Response · Authors · 2024-12-03
> >
> > Dear reviewer aWZ5, We'd appreciate any follow-up questions and your revised review soon.

---

### Official Review · Reviewer_oJPV · 2024-11-04

**Soundness:** 3
**Presentation:** 3
**Contribution:** 2
**Rating:** 5
**Confidence:** 3

**Summary:**

The authors present a self-supervised framework for leveraging contrastive learning to model cell state dynamics from time-lapse imaging.
The model allows temporally adjacent states to be mapped closely together helping achieve accurate, efficient, and label-free analysis of dynamic cell states under perturbations like viral infection. The paper presents a unique application of contrastive learning in cellular imaging and stands out for its temporal coherence and robustness in infection state classification.

**Strengths:**

- The use of temporally-aware contrastive learning enables efficient modeling of time-dependent cellular changes, showing improved performance over traditional contrastive methods, especially under perturbative conditions.
- The framework facilitates rapid annotation of cell states, potentially decreasing the reliance on human-intensive and subjective labeling, a significant advancement over prior approaches.
- By using a cell-aware approach, the framework attempts to address the intrinsic heterogeneity across cell populations, an advantage over traditional time-agnostic contrastive approaches.

**Weaknesses:**

- The choice of a 30-minute interval as the temporal offset might not generalize across other biological systems with different dynamics, limiting the model adaptability.
- The reliance on phase and fluorescence imaging could constrain its utility where alternate modalities are necessary.
- Since cell-aware and time-aware sampling use specific tracked cells, the embeddings may risk overfitting to individual cell trajectories instead of generalized dynamics.
- Although contrastive learning was chosen, the paper lacks in-depth comparisons with generative methods that authors summarize in the related work.
- By setting a fixed temporal offset, the model may miss capturing events that unfold asynchronously or at variable rates in different cells.
- Models relying on phase channels for cell division detection may struggle with subtler morphological changes that require fluorescence markers.

**Questions:**

- How does the model perform for other tasks beyond infection classification? Like, for example, tracking mitotic spindle dynamics during first cell division in embryonic development?

- How does the model handle potential noise or artifacts in the time-lapse imaging data? How are hyperparameters tuned and how sensitive is the model to the choice of these hyperparameters?

---

> ### Author Response · Authors · 2024-11-26
>
> Thank you for the review and for acknowledging the strength of our work. We address your feedback below:
>
> **Weaknesses:**
> ```
> The choice of a 30-minute interval as the temporal offset might not generalize across other biological systems with different dynamics, limiting the model’s adaptability.
> ```
> We address this concern by doing new computational experiments: we use public ALFI data to demonstrate that models trained with multiple time intervals (7 minutes - 90 minutes) reliably embed cell cycle dynamics in the test dataset acquired with 7-minute intervals (Figure 2 and corresponding appendix figures). Interestingly, we observe that when the time interval of contrastive sampling approaches the typical time interval of morphological change,  the dynamic range of the embeddings is maximized.   We also demonstrate that infected A549 cells imaged with a 10-minute time resolution can be reliably classified using an embedding model trained with cells imaged with a 30-minute time resolution (Figure 3, panels e and f).
> ```
> The reliance on phase and fluorescence imaging could constrain its utility where alternate modalities are necessary.
> ```
> Phase and fluorescence modalities are broadly used in drug discovery and cell biology. The concepts we present can be extended to other time-lapse datasets. We have demonstrated the model's applicability with multiple phase and fluorescence modalities, particularly using the publicly available DIC dataset, ALFI.
>
> ```
> Since cell-aware and time-aware sampling use specific tracked cells, the embeddings may risk overfitting to individual cell trajectories instead of generalized dynamics.
> ```
> Our sampling strategy promotes intra-track smoothness and inter-track discrimination and learns generalized dynamics, e.g., propagation of infection in a population (Figure 3, panel f). In the revised Figure 3, we have included embeddings of multiple independent test datasets from different cell types, imaging modalities, microscope systems, temporal sampling, and fluorescence markers to show that the dynaCLR models generalize across various experimental conditions.
>
> ```
> Although contrastive learning was chosen, the paper lacks in-depth comparisons with generative methods that authors summarize in the related work.
> ```
> In the revision, we compare this work with published work on time-regularized generative modeling (https://www.molbiolcell.org/doi/full/10.1091/mbc.E21-11-0561). Thank you for the suggestion. A thorough comparison with generative models is out of the scope of this work. We think a thorough evaluation of generative and contrastive models of time-lapse data is a great topic for future work.
>
> ```
> By setting a fixed temporal offset, the model may miss capturing events that unfold asynchronously or at variable rates in different cells.
> ```
> We have demonstrated the capture of stages of cell cycle using ALFI, a publically available dataset, in which the mitosis spans over multiple time points and unfolds asynchronously. We validated the results by comparing them with available human annotations. Please see Figure 3 to see the generalization of the model enabling the capture of mitosis in a cell type that was not used to train the models.
> ```
> Models relying on phase channels for cell division detection may struggle with subtler morphological changes that require fluorescence markers.
> ```
> We'd appreciate the elaboration of this question. We agree that subtle changes in morphology may not be captured by any single channel. That is why we are proposing a flexible method that enables the embedding of 3D multi-channel datasets.  Specifically, we are leveraging fluorescent labels for organelle phenotyping, see Figure 3, panels g, h, and I.
>
> **Questions:**
> ```
> How does the model perform for other tasks beyond infection classification? Like, for example, tracking mitotic spindle dynamics during first cell division in embryonic development?
> ```
> We now show that the framework can be used to learn phenotypes across multiple datasets and microscopes to classify cell division, classify infection, and learn organelle responses to infection.
> ```
> How does the model handle potential noise or artifacts in the time-lapse imaging data? How are hyperparameters tuned and how sensitive is the model to the choice of these hyperparameters?
> ```
> In Figure 3, we demonstrate the model's generalizability to different microscopes by performing prediction on an independent dataset with differences in noise and imaging parameters. The new data in Figure 2, Figure 3, and Table 1 illustrate that dynaCLR models lead to useable embeddings for a larger range of time interval hyperparameters.

---

> > ### Author Response · Authors · 2024-12-03
> >
> > Dear reviewer oJPV,
> > We'd appreciate any follow-up questions and a revised review soon.

---

> > > ### Comment · Reviewer_oJPV · 2024-12-03
> > >
> > > I thank authors for detailed response to reviewer's comments. Based on the quality of the revisions and the responses, I will not adjust my previous evaluation and maintain my current rating.

---

### Author Response · Authors · 2024-11-26
**Summary of the revision**

We want to thank all the reviewers for their detailed and constructive feedback and look forward to an active discussion over the next few days.

The reviewers recognized the timeliness of the problem we are addressing and the rigor of our solution.

___“ temporally-aware contrastive learning enables efficient modeling of time-dependent cellular changes”, “framework facilitates rapid annotation of cell states, potentially decreasing the reliance on human-intensive and subjective labeling, a significant advancement over prior approaches”___  (reviewer oJPV);

___“The learned representation was used in several dowstream tasks.”  (reviewer: aWZ5)___

___“The addition of generalization experiments (though only on one additional set of data) provides some grounding to the results – without this section I don’t think the paper has much weight. I would encourage the authors to extend this to more datasets if possible.”___ (reviewer: UA5v).

The key constructive feedback from reviewers concerns the method's novelty and generalizability. We have added new results aligned with the paper's original scope on the generalized embedding models trained with public and in-house datasets. We respond to the reviewers' feedback individually. Below is a summary of the main changes to the structure and content of the paper.

* The main figures and appendix figures report the following data :
   * Fig 1: Overview of dynaCLR framework.
      - Appendix Fig 1: napari plugin for interactive annotation of cell states in embedding space.
   * Fig 2: Evaluation of time-aware contrastive sampling using public dataset (ALFI)
      - Appendix Fig 2: Displacement, dynamic range, and smoothness of cell tracks in the embedding as a function of the contrastive sampling strategy.
      - Appendix Fig 3: PHATE visualizations of embeddings with contrastive sampling strategies.
      - Appendix Fig 4: Evaluation of contrastive loss using in-house (cellular response to infection) data.
      - Appendix Fig 5: Number of significant principal components of embeddings as a function of contrastive sampling strategy.
   * Fig. 3: Evaluation of generalization of the learned embeddings across cell types and microscopes using public (ALFI) and in-house (cell and organelle response to infection) data.
      - Appendix Fig 6: robustness of dynaCLR embeddings to tracking errors in the data.
      - Appdneix Fig 7: detecting cell division events in infected cells.
      - Appendix Fig 8: visual inspection of remodeling of endoplasmic reticulum due to infection.
   * Fig 4: Explanation of learned embeddings with engineered features and occlusion-based class attribution.


* We report computational experiments to show that the time-interval hyperparameter can be changed to tune embeddings' dynamic range and smoothness (Fig. 2 and Table 1).

* In an exciting development, we find that the dynaCLR embedding model generalizes across unseen microscopes and contrast methods (Fig. 3). We think that the self-supervised learning of temporally smooth representations of 4D (3D, multi-channel) biological datasets is a valuable novel contribution of this work.

* NT-Xent and triplet loss lead to embeddings with similar structures  (Appendix Fig 4), and time-aware contrastive sampling improves the dynamic range and smoothness of embeddings independent of the loss function.

* We improve the writing throughout to highlight novel methodological aspects of the work, namely,
  * The first use of time-aware contrastive sampling to learn embeddings from 4D tensors.
  * Application to diverse biological datasets in which cell states are labeled by a human or via an experimental fluorescent marker.

* We now use precise mathematical notation to clarify the contrastive sampling strategy, the metric of dynamic range in embedding space, and the metric of smoothness in embedding space.

---

### Meta-Review · Area_Chair_wvze · 2024-12-20

**Metareview:**

The paper describes a self-supervised framework for modeling cell and organelle dynamics in time-lapse microscopy datasets called dynaCLR. The reviewers unanimously recommend rejection, citing a lack of significant methodological novelty and a lack of sufficient rigor in presentation from a machine learning paper.

**Additional Comments On Reviewer Discussion:**

While there was not significant reviewer discussion, largely due to the overwhelmingly negative initial scoring of the paper, one reviewer responded that the revision did not sufficiently improve their evaluation of the paper.

---

### Decision · Program_Chairs · 2025-01-22

Reject